

# Nicotinamide riboside exerts protective effect against aging-induced NAFLD-like hepatic dysfunction in mice

Xue Han[1,*], Xiaogang Bao[2,*], Qi Lou[1], Xian Xie[3], Meng Zhang[4], Shasang Zhou[1], Honggang Guo[1], Guojun Jiang[5] and Qiaojuan Shi[1]

[1] Laboratory Animal Center, Zhejiang Academy of Medical Sciences, Hangzhou, Zhejiang Province, China
[2] Department of Orthopedic Surgery, Spine Center, Changzheng Hospital, Second Military Medical University, Shanghai, China
[3] Hospital of Stomatology, Zhejiang University School of Medicine, Hangzhou, Zhejiang Province, China
[4] Department of Clinical Laboratory, Tongde Hospital of Zhejiang Province, Hangzhou, Zhejiang Province, China
[5] Zhejiang Xiaoshan Hospital, Hangzhou, Zhejiang Province, China
[*] These authors contributed equally to this work.

Corresponding authors
Guojun Jiang, jguojun999@163.com
Qiaojuan Shi, shiqiaojuan@163.com

## ABSTRACT

**Background & Aims**. Aging is one of the risk factors of non-alcoholic fatty liver disease (NAFLD). Yet, the mechanism underlying the aging-associated NAFLD-like syndrome is not fully understood. Nicotinamide adenine dinucleotide (NAD), a ubiquitous coenzyme, has protective effects against aging. Here, we investigated the actions of NAD precursors nicotinamide riboside (NR) on the development of aging-induced NAFLD.

**Methods**. NR supplemented food (2.5 g/kg food) was applied to aged mice for three months while normal chow to the other groups. Body weight, food intake, liver weight and fat pat mass were measured. The serum concentrations of lipid content, alanine aminotransferase (ALT), aspartate aminotransferase (AST) and NAD were determined by biochemical assays. Pathological assessment and immunohistochemistry analysis of hepatic tissues were used to evaluate the effect of NR on NAFLD development and inflammatory infiltration.

**Results**. NR repletion significantly reduced fat pat mass in aged mice, while not altered the body weight, food intake, and liver weight. NR repletion significantly rescued the NAD reduction in aged mice. The total cholesterol and triglyceride levels could be lowered by NR repletion in aged mice. The AST level was also significantly reduced by NR repletion in aged group, while the ALT level lowered but without significance. Notably, moderate NAFLD phenotypes, including steatosis and hepatic fibrosis could be markedly corrected by NR repletion. In addition, Kupffer cells accumulated and inflammatory infiltration could also be remarkably reversed by NR repletion in aged mice.

**Conclusion**. Aging was associated with NAFLD-like phenotypes in mice, which could be reversed by oral NR repletion. Therefore, oral NR uptake might be a promising strategy to halt the progression of NAFLD.

## INTRODUCTION

Non-alcoholic fatty liver disease (NAFLD), is a metabolic disorder characterized by imbalanced lipid metabolism and fatty acid accumulation in the liver. And this complex condition involves the progression, from simple steatosis to inflammation (nonalcoholic steatohepatitis), and then to severe fibrosis and hepatocellular carcinoma, which are the major predictors of death in patients with NAFLD (*Romeo, 2019*). Besides, NAFLD is widely considered as hepatic manifestation of metabolic syndrome. NAFLD is associated with type 2 diabetes mellitus, obesity and insulin resistance, which are the main features of metabolic syndrome (*Wan et al., 2016*).

Aging is a physiological process associated with multiple organ function decline, especially the liver. Specifically, aging could trigger hepatic steatosis and progressive inflammation (*Jadeja et al., 2019*). It was found that the prevalence of the NAFLD increases markedly with older age (*Lee et al., 2007*; *Amarapurkar et al., 2007*). A clinical research, involving 589 consecutive liver biopsies, reveals that age over 30 is an independent risk factors for liver steatosis (*Lee et al., 2007*). Morbidity and age-adjusted mortality of chronic liver diseases are often more severe in older people (*Frith, Jones & Newton, 2009*). The aging-related liver pathology includes hepatic morphological disorder, hepatocyte polyploidization, and the reduced mitochondrial density (*Gan, Chitturi & Farrell, 2011*; *Wu et al., 2019*). However, the mechanisms that underlie aging-related hepatic dysfunction have not been fully elucidated. Therefore, investigation of the mechanism and exploration of the therapeutic strategies are vital for the disease management.

It is well established that aging process is featured in mitochondria dysfunction and resulted depletion of nicotinamide adenine dinucleotide (NAD) depletion (*Kang et al., 2013*; *Gomes et al., 2013*; *Andreani et al., 2018*). NAD is involved in many cellular functions, and plays a crucial role in energy metabolism (*Watroba et al., 2017*). The salvaging synthesis pathway is predominant in NAD synthesis in mammalian cells. Nicotinamide riboside (NR) is presumed as a NAD precursor for this salvaging pathway (*Moon, Kim & Shin, 2018*). Once it enters the cell, NR is converted to nicotinamide mononucleotide (NMN) by rate-limiting enzyme nicotinamide phosphoribosyltransferase (NAMPT). Then, NMN is further metabolized to NAD. Additionally, the de novo biosynthesis of NAD from trytophan is considered to be another pathway for NAD synthesis. However, the de novo biosynthesis pathway only occurred in limited cell types (*Yoshino, Baur & Imai, 2018*). NAD level reduction and unbalance between its synthesis and consumption is proved to be related to aging-associated diseases such as Parkinson's disease (PD) (*Ješko et al., 2017*) and Alzheimer's disease (AD) (*Xie et al., 2019*). Hence, NAD supplementation might be a promising strategy to restore the cellular function. Better yet, NR is found in milk, making it possible through dietary modulation (*Bieganowski & Brenner, 2004*). In contrast to NR, other precursors of NAD biosynthesis, such as nicotinic acid (NA), nicotinamide or NMN, have been shown to be severe flushing or toxin in pre-clinical trials (*Bogan & Brenner, 2008*; *Di Stefano et al., 2015*). Therefore, NR was highlighted as a promising compound for the rescue of NAD level. Notably, NAD complementation has been previously reported to be beneficial in other conditions. *Karpe & Frayn (2004)* showed that NAD complementation

improved the plasma lipid and cholesterol profiles, and ameliorated metabolic disorder. Modulation of the cellular NAD level can attenuate hepatic steatosis and inflammation in mice fed with methionine-choline-deficient diet (*Katsyuba et al., 2018*). Evidences also display that replenish NR has beneficial effects on NAFLD, insulin sensitivity and AD in mice (*Gariani et al., 2016*; *Cantó et al., 2012*; *Xie et al., 2019*). In short, NAD has pronounced effects on hepatic homeostasis.

However, the relation between NAD complementation through NR supplementation and aging-associated hepatic dysfunction has not been investigated. In the current study, we fed the aging mice with NR supplied food for 3 months. We aimed to explore the role of NR repletion on aging-induced NAFLD and inflammatory infiltration in aged mice.

## MATERIALS & METHODS

### Animals

Female C57BL/6J mice in 3-month old and 14-month old were purchased from the Laboratory Animal Center of Zhejiang Province (Hangzhou, China), and used as young mice, aged mice and NR supplied aged mice respectively. All mice were maintained in an environmentally-controlled room (12 h light-dark cycle, 20–26 °C, relative humidity 50%), fed a standard chow with free access to water. All animal experiments were performed in accordance with the National Institutes of Health Guide for the Care and Use of Laboratory Animals. Procedures were approved by the Institutional Animal Care and Use Committee of the Laboratory Animal Center of Zhejiang Province and the experimental protocols were approved by the Ethics Committee of Laboratory Animal Care and Welfare, Zhejiang Academy Medical Sciences, with the proved number DLSY.2017(80).

### NR supplementation

The mice were divided to three groups: (1) Young mice (Young); (2) Aged mice (Aged); and (3) NR treated aged mice (Aged + NR). NR (Baikai Chemical Technology Co., Ltd, Hangzhou, China) was mixed into the pellets with the concentration of 2.5 g/kg (Zhejiang Academy of Medical Sciences, Hangzhou, China). Food consumption of aged mice for ten days were measured and the average food intake was estimated at 160 g/kg. According to the food intake, the aged mice orally treated with NR around 400 mg/kg/day. The food containing NR was supplied from fifteen-month old and four-month old for C57BL/6J mice and lasted for three months until sacrificed. Mice in the Young and Age groups were received common food correspondingly. After 3 months, mice were used for *ex vivo* studies.

### Body weight and food intake determinations

Body weight changes of each group were measured at the end of NR administration. Additionally, the food intake of aged mice was measured in cages every 3–5 days. The average daily amount of each mouse was calculated.

### Liver weight and fat pat mass measurements

Mice of three groups were euthanized by chloral hydrate (800 mg/kg) injection intraperitoneally, after an overnight fasting period. Mice were transcardially perfused

with 4 °C saline. The livers and total fat pat were quickly removed, carefully cleaned, and blotted dry. Then the collected samples were measured carefully. The ratios between tissue mass and body weight were calculated respectively.

## Blood biochemical assays

Blood samples were acquired from inferior vena cava of anesthetized mice. About 0.7 ml blood was harvested. Samples were still standing at room temperature for 40 min and at 4 °C for 2 h, then followed by 3,000 rpm centrifugation for 10 min. The supernatant was used for blood biochemical index measurement. According to the manufacturer's instructions, triglyceride(TG), total cholesterol (TC), alanine aminotransferase (ALT) and aspartate aminotransferase (AST) contents in serum were quantified by an automatic biochemistry analyzer (Backman). Blood NAD concentration was determined by a commercial NAD Quantitation Colorimetric Kit (K337-100; Biovision, San Francisco, CA, USA).

## Histological analysis

Those isolated livers, taken from the same lobe, were fixed in 4% paraformaldehyde for 8 h and then were embedded in paraffin for histological processing. Samples were cut into thin section (five µm) and stained with hematoxylin and eosin (H&E) to assess histopathology, and Masson's trichrome for collagen evaluations. Images were obtained at 200 magnifications under the inverted phase-contrast microscope (Leica Microsystems, Wetzlar, Germany).

Scoring for steatosis (severity and extension) was performed in a blinded and independent method by two observers as described before (*Gariani et al., 2016*). The analysis used a scale of 0–4, where 0 referred to absent of vacuolation in the liver, 1 referred to 2 or 3 vacuoles per hepatic cord per lobule, 2 referred to less than 50% of the lobule has fatty vacuolation, 3 referred to more than 50% of the lobule has fatty vacuolation, and 4 corresponded to nearly the entire lobule has fatty infiltration. Moreover, the focal extention was referred to 1, multifocal was referred to 2, and almost total diffuse was referred to 3.

## Immunohistochemical analysis

For immunohistochemical analysis, thin sections blocked by 5% goat serum followed by incubating in specific primary antibodies. PBS was applied to wash the sections three times. Then, samples was stained with horseradish peroxidase-conjugated secondary antibodies and visualized by substrate DAB. Images were taken with a microscope (Leica, 200×) under same acquisition settings for each section. The primary antibodies were used as follow: TGF-β (MAB240-100, R&D System, 1:600 dilution), F4/80 (LS-C96373-100, Lifespan, 1:1000 dilution), CD68 (ab125212, Abcam, 1:600 dilution), IL-1β (SRP8033, Sigma, 1:1000 dilution), TNF-α (ab6671, Abcam, 1: 600 dilution).

## Statistical analyses

Data were expressed as mean ± SEM. Values from different groups were analyzed using one-way ANOVA followed by Newman–Keuls multiple comparison test. Statistical analysis was done in GraphPad Software (Prism Version 5.01). Statistical significance was considered as $P < 0.05$.

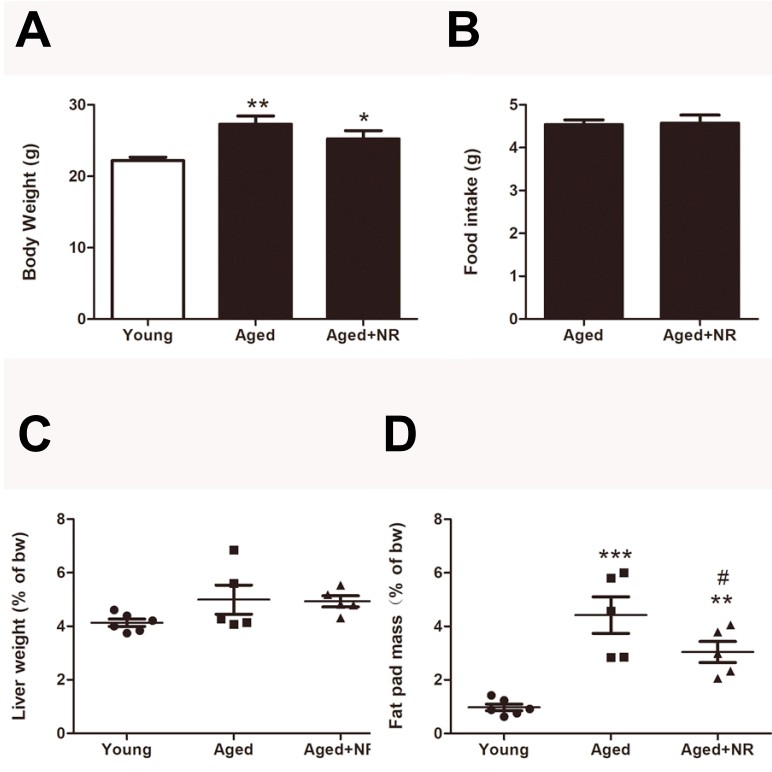

**Figure 1** **Effects of NR repletion on the body weight, food intake, relative liver weight and fat pad mass of aged mice.** Compared with the young, aged C57BL/6 mice were treated with NR (400 mg/kg/day) for 3 consecutive months. (A) Effect of NR repletion on the body weight of aged mice. (B) Effect of NR supplementation on the food intake of aged mice. Changes of relative liver weight (C) and fat pat mass (D) caused by NR treatment. Values are mean $\pm$ SEM, ($n = 5$–6 per group). *$P < 0.05$, **$P < 0.01$, ***$P < 0.001$ *vs* Young mice, #$P < 0.05$ *vs* aged mice.

# RESULTS

## Changes of body weight, food intake, relative liver weight and fat pat mass in NR treated aged mice

After 3 months of NR supplied, body weight in aged mice with NR repletion was a little lower than the aged mice ($25.2 \pm 1.2$ g *vs* $27.3 \pm 1.1$ g, $P > 0.05$), although no significant difference was shown (Fig. 1A). There was also no significant difference in food intake or liver to body weight ratio between NR supplied aged mice and aged mice. (Figs. 1B and 1C). As shown in Fig. 1D, ageing was sufficient to induce fat pat mass to body weight ratio increased compared with young. While the ratio was greatly decreased in aged mice with NR repletion when compared to the aged mice ($3.0 \pm 0.4\%$ *vs* $4.4 \pm 0.7\%$, $P < 0.05$).

## NR favoured lipid homeostasis and hepatic steatosis in aged mice

To answer whether the NR supplied could improve the susceptibility to development of NAFLD in aged mice, we used 15 months old mice with 3 months feeding of NR. As a result, serum elevation in AST and ALT of aged mice indicated an impaired of liver. While the level of AST was greatly attenuated with NR ($98.8 \pm 8.56$ U/L vs $124.7 \pm 10.5$ U/L, $P < 0.05$,

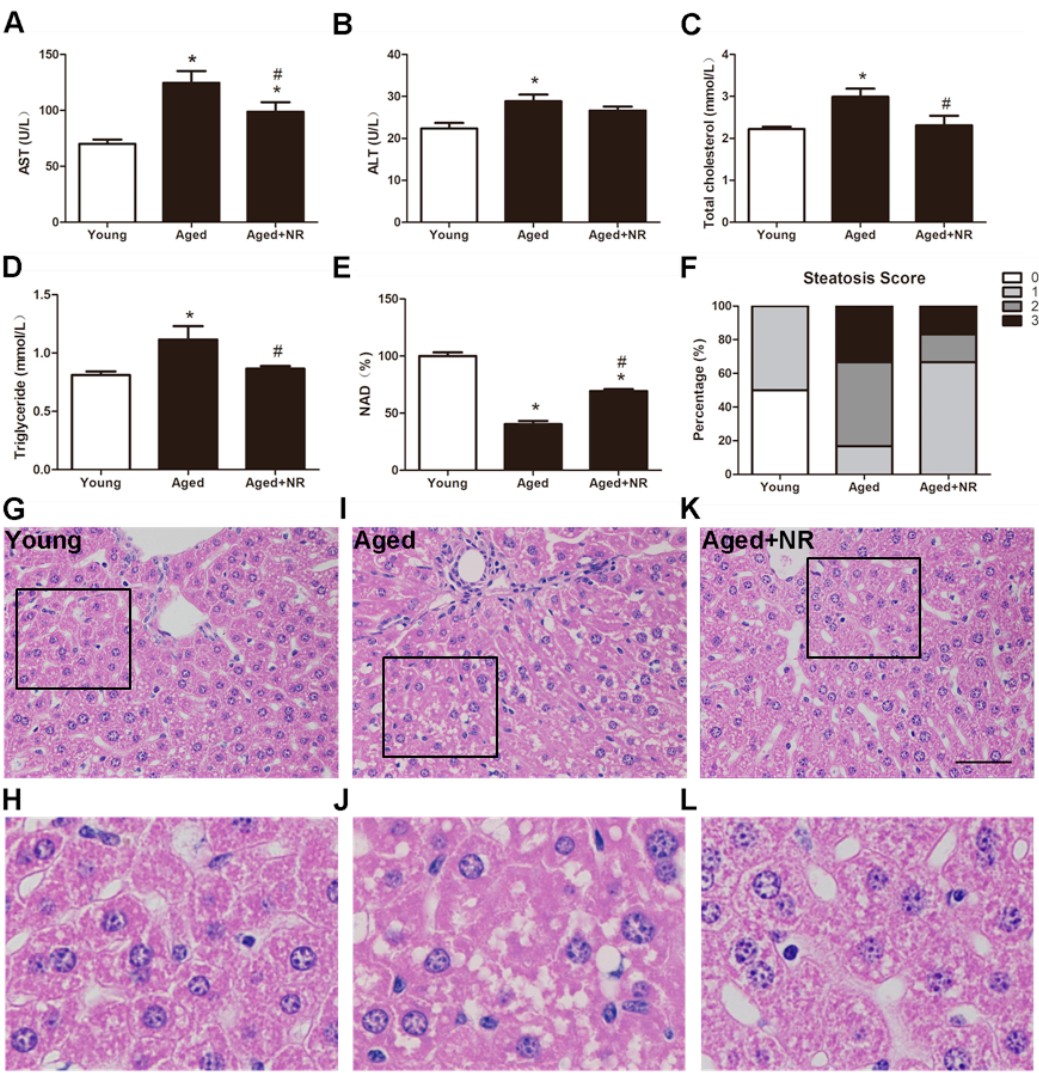

**Figure 2  Effect of NR repletion on the development of the NAFLD in aged mice.** The levels of AST (A), ALT (B), total cholesterol (C), triglyceride (D), and NAD (E) in serum of aged mice. (F) hepatic steatosis was reduced by NR administration in aged mice. The percentage of classified livers in each of the four steatosis categories in different groups was as follows: 0, no vacuolation; 1, 2 or 3 vacuoles; 2, less than 50% of fatty vacuolation; 3, more than 50% of fatty vacuolation. (G–L) Representative images stained with H&E of liver tissues of aged mice (400×; scale bar, 50 µm); box regions are shown at higher magnification under the original pictures. Values are mean ± SEM, ($n$ = 5–6 per group). $^*P < 0.05$ *vs* Young mice, $^\#P < 0.05$ *vs* aged mice.

Fig. 2A) without ALT (Fig. 2B). The TC and TG contents were significant elevated in aged mice when compared to young mice. NR supplied aged mice significantly reduced both TC (2.31 ± 0.23 mmol/L vs 2.99 ± 0.19 mmol/L, $P < 0.05$, Fig. 2C) and TG level (0.87 ± 0.02 mmol/L vs 1.12 ± 0.12 mmol/L, $P < 0.05$, Fig. 2D). The results were in agreement with improved fat accumulation, suggesting NR protects against age-induced lipid disorders and matched by changes in NAD concentration (Fig. 2E). Moreover, H&E staining presented

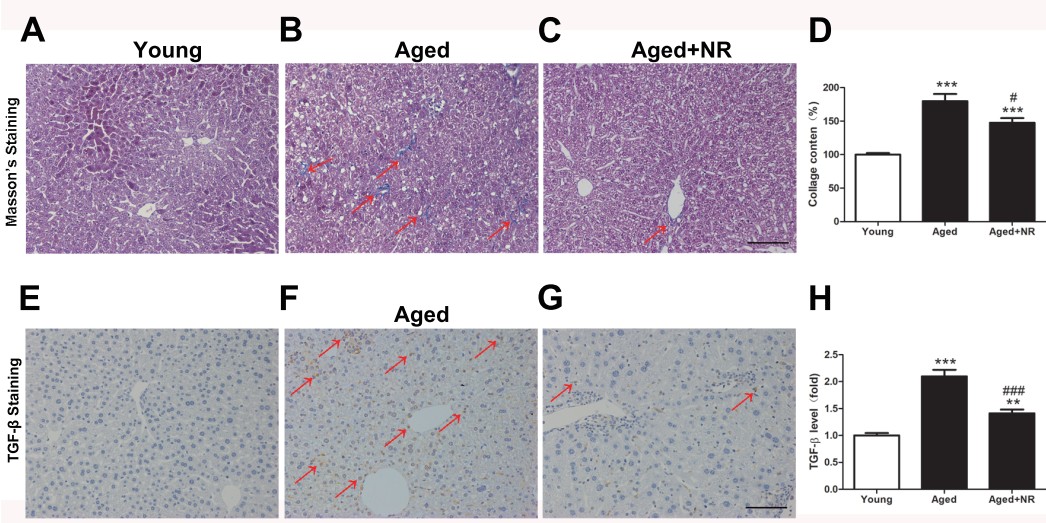

**Figure 3  Effect of NR repletion on liver fibrosis in aging-relative NAFLD model.** (A–C) Masson's staining in livers from aged mice and (D) quantitative analysis. Red arrows show positive blue staining for Masson. (E–G) Protein level of TGF-β was detected by immunohistochemistry and (H) quantittive analysis. Red arrows show positive brown staining for TGF-β. 200×; Scale bars, 100 μm. Values are mean ± SEM, ($n = 5$–6 per group). **$P < 0.01$, ***$P < 0.001$ *vs* Young mice, #$P < 0.05$, ###$P < 0.001$ *vs* aged mice.

hepatocellar irregularity shaped and severity of steatosis in aged mice (Figs. 2G–2L). The histology score in aged mice was greatly improved by NR supplied (Fig. 2F). These observations suggested that aging could promote the lipid accumulation and the ensuing development of NAFLD-like hepatic dysfunction. Also, NR was demonstrated to prevent age-induced hepatic steatosis.

## NR improved hepatic fibrosis in aged mice

To further determined the influence of NR on development of NAFLD in aged mice, Masson's trichrome and TGF-β staining were performed. NR weaken hepatic collagen and fibrosis, as revealed by less Masson's trichrome staining and TGF-β staining (Figs. 3A–3D, 3E–3H).

## NR alleviated inflammation infiltrated in liver of aged mice

We investigated the influence of NR on hepatic inflammation in aged mice. Immunohistochemical staining for F4/80 and CD68 indicated that accumulated Kupffer cells were obviously reduced in liver of NR supplemented aged mice compared with aged mice (Figs. 4A–4D, 4I). In agreement with NR-induced improvement in macrophagocyte infiltration, there was also a significant down-regulation of pro-inflammatory cytokines IL-1β and TNF-α expression in liver from NR supplied aged mice compared with aged mice (Figs. 4E–4H, 4J).

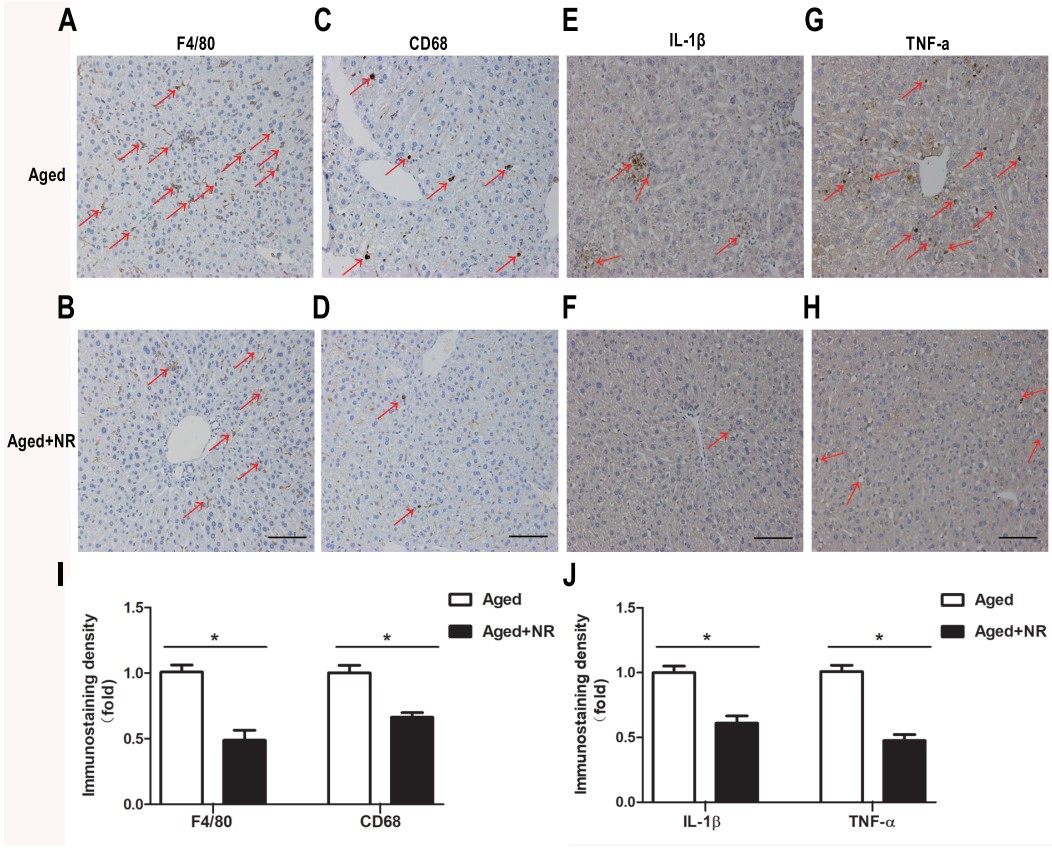

**Figure 4** **Effect of NR repletion on inflammatory infiltration of aging-relative NAFLD model.** (A–D) Immunohistochemistry staining in liver of aged mice showed the effect of NR supplementation on Kupffer cell accumulation and (I) quantittive analysis. Red arrows show positive brown staining for F4/80 or CD68. (E–H) Protein levels of IL-1β and TNF-α were detected by immunohistochemistry and (J) quantittive analysis. Red arrows show positive brown staining for IL-1β or TNF-α. 200×; Scale bars, 100 μm. Values are mean ± SEM, ($n = 5$–6 per group). *$P < 0.05$.

## DISCUSSION

The major findings of the present study were as follows. First, the supplementation of NR ameliorated lipid homeostasis and hepatic steatosis in aged mice. Second, the supplementation of NR reduced collagen deposition and hepatic fibrosis in liver from aged mice. Finally, we showed that NR treatment decreased Kupffer cells infiltrating as well as lowered IL-1β and TNF-α expression in liver from aged mice.

Prevalence of NAFLD increases dramatically with age, although this disease appears in different age groups (*Zhou et al., 2019*). There are strong evidences suggesting steatohepatitis and fibrosis are associated with aging, which results in a higher mortality in elderly individuals with NAFLD (*Argo et al., 2009*; *Ooi et al., 2018*). Researchers have identified several mechanisms underlying the age promotion the morbidity of NAFLD. Physiological changes characterize aging may trigger the development of components of the metabolic disturbance. For example, the functional decrease in the lysosomal

degradative pathway of autophagy appears to be remarkable in aged individual, which may encourage lipid accumulation in the liver (*Martinez-Lopez, Athonvarangkul & Singh, 2015*; *Chi et al., 2019*). Furthermore, the level of oxidative stress, inflammation and DNA damage increase with aging, and these excessive elevations have also been implicated as mediators of NAFLD pathogenesis. A previous study has reported that the histological grade of steatosis similarly increased in aged mice compared with young and middle mice (*Fontana et al., 2013*). Recently, the deteriorate morphology and function of livers have also been observed in natural aging rat models (*Minhas et al., 2019*). In this study, our results indicated that 18-month-old C57BL/6J mice exhibited an impaired lipid homeostasis including body weight gain, fat accumulation and serum TG and TC increase. These mice showed great susceptibility to development of NAFLD, reflected in steatosis with moderate fatty infiltration. Liver is a vital regulator of metabolism. Therefore, it is important to maintain hepatic function of elderly. Nevertheless, the little data are currently available in molecular mechanism for aging-related NAFLD.

NAD is a substrate for multiple enzymes of sirtuin family and participates in multiple cellular functions, including DNA repair, energy metabolism, and regulation the activity of the sirtuins by transcriptional control (*Hoxhaj et al., 2019*). It is well-established that aging and fatty liver related dysfunction leads to a pronounced effect on decline of NAD concentration in liver. *Fan et al. (2018)* have reported that the expression of hepatic mRNA of regulating NAD biosynthesis is greatly reduced in aged mice or challenged high-fat diet (HFD) mice. Additionally, this phenomenon seems to be a toxic element, providing destructive actions because a shortage of NAD links ageing to progressive liver damage. The beneficial effects of NAD regiment on fatty liver have been reported, for instance in a liver-specific Sirt1 knockout mouse (*Katsyuba et al., 2018*) and in an enzyme-dead NAMPT transgenic mouse (*Zhou et al., 2016*). Thus, supplementation NAD pool may be an attractive therapy strategy for liver damage related diseases in elderly individual. Notably, NR, this vitamin B3 analog, as a precursor of NAD biosynthesis, is commonly used to boost NAD pool (*Jiang et al., 2019*). Here we showed that the replenishing of NR has beneficial effect on liver of aged mice. Our study demonstrated that aged mice administrated of NR (400 mg/kg/day) for 3 months improved lipid disordered. Moreover, NR treatment exhibited an amelioration in hepatic steatosis and fibrosis that was matched by an augmented blood NAD concentration, implying a systemic NAD replenishing in aged mice.

An extensive body of evidence indicates that chronic inflammation contributes to the degenerative changes of full-length tissues in the context of aging. Even normal brain of aged individual is characterized by increased inflammation and subsequently elevated pro-inflammatory cytokines (*Frank et al., 2006*). As inflammation rose by age shows a reduction in adequate NAD content in brain of the murine (*Braidy et al., 2011*). Previous study has also demonstrated that genetic blockade of NAD synthesis exerts inflammatory effects on the liver reflecting by activation NLRP-3 inflammasome pathway and production of IL-18 and IL-1β (*Jiang et al., 2019*). The other independent group shows that pharmacological inhibition of de novo NAD synthesis strengthens transcription genes involved in inflammation, including Desmin and Tgfb (*Katsyuba et al., 2018*). Intriguingly,

increasing the NAD concentration leads to promote pro-inflammatory cytokine synthesis by activated immune cells (*Van Gool et al., 2009*). Consistently, decreasing the NAD pool causes innate immune disorder in aging-associated diseases (*Minhas et al., 2019*). In the present study, we found that the supplementation of NR obviously weakened Kupffer cell accumulation accompanied by inhibiting expression of IL-1β and TNF-α. In the context of lipid accumulation, macrophages are recruited into liver and pro-inflammatory cytokines subsequently produced in liver of aged mice. This data is probably discrepanct with several previous reports, thus further evidences are still needed to confirm the relationship between NAD level and inflammatory reaction.

## CONCLUSIONS

In summary, in the present study we show that aging-related NAD deficiency causes pathologic changes and inflammation infiltration in liver of aged mice. The replenishment of NAD, by treated with NR, is able to protect against aging-induced hepatic steatosis, which is possibly associated with an improvement in reduction of pro-inflammatory cytokines, such as IL-1β and TNF-$\alpha$. Our study raises the possibility of NR to alleviate NAFLD-like liver injure in aged individuals, suggesting the clinical advantage of NR during vitamin supplementation therapy. Further investigations are warranted to treat aging-related liver diseases by NAD supplementation strategy.

## ACKNOWLEDGEMENTS

The authors are grateful to Ms Zhang and Ms Lu (Department of Pharmacology, Key Laboratory of Medical Neurobiology of Ministry of Health of China, Zhejiang University School of Medicine, Hangzhou, China) for their technical assistance.

### Funding
This study was supported by the Natural Science Foundation of Zhejiang (Grants LGJ18H310002, LQY19H090001, LY18H310009 and LQY18C040001) and the Shanghai Committee of Science and Technology, China (Grant No. 15411951000 and 2018QN13). The funders had no role in study design, data collection and analysis, decision to publish, or preparation of the manuscript.

### Grant Disclosures
The following grant information was disclosed by the authors:
Natural Science Foundation of Zhejiang: LGJ18H310002, LQY19H090001, LY18H310009, LQY18C040001.
Shanghai Committee of Science and Technology, China: 15411951000, 2018QN13.

### Competing Interests
The authors declare there are no competing interests.

## Author Contributions

- Xue Han conceived and designed the experiments, performed the experiments, analyzed the data, prepared figures and/or tables, authored or reviewed drafts of the paper, approved the final draft.
- Xiaogang Bao, Meng Zhang and Shasang Zhou performed the experiments.
- Qi Lou performed the experiments, contributed reagents/materials/analysis tools.
- Xian Xie analyzed the data, contributed reagents/materials/analysis tools.
- Honggang Guo analyzed the data, contributed reagents/materials/analysis tools, prepared figures and/or tables.
- Guojun Jiang conceived and designed the experiments, contributed reagents/materials/analysis tools, authored or reviewed drafts of the paper.
- Qiaojuan Shi conceived and designed the experiments, prepared figures and/or tables, authored or reviewed drafts of the paper, approved the final draft.

## Animal Ethics

The following information was supplied relating to ethical approvals (i.e., approving body and any reference numbers):

The Institutional Animal Care and Use Committee of the Laboratory Animal Center of Zhejiang Province provided full approval for this research (DLSY.2017(80)).

## Data Availability

The raw measurements are available in the Supplemental File.

## Supplemental Information

Supplemental information for this article can be found online at http://dx.doi.org/10.7717/peerj.7568#supplemental-information.

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
