# Peer review of "Nicotinamide riboside exerts protective effect against aging-induced NAFLD-like hepatic dysfunction in mice"

_PeerJ, doi:10.7717/peerj.7568_

## Round 0.1 · original submission · Minor Revisions

· Academic Editor

Minor Revisions

Based on the advice received, I have decided that your manuscript could be reconsidered for publication should you be prepared to incorporate minor revisions. However, we are not prepared to accept your manuscript in its present form.


·

Basic reporting

No comments.

Experimental design

No comments.

Validity of the findings

No comments.

Additional comments

In their manuscript, Han and colleagues report the effects of nicotinamide riboside, a nicotinamide adenine dinucleotide precursor, on development of age-induced non-alcoholic fatty liver disease and inflammation liver in mice. Identifying novel, effective therapeutic compounds for liver diseases is an important goal, and the paradigm used in the present study is interesting to investigate potential new therapeutics. The experiments seem to be well performed and the results are sound, but some points need to be addressed by the authors.

1. In the keywords section, please do not use abbreviations;
2. Overall, there are quite a few grammatical errors along with sentences that are difficult to interpret. This manuscript would benefit greatly from additional proofreading and editing; For example: the word “that” sometimes do not necessary (ex. first phrase od introduction section).
3. The introduction section has much information, it is good and interesting, but the introduction needs a connection to improve the read;
4. The introduction could use additional background information about the role of NR in other diseases;
5. The last paragraph of introduction section, need demonstrate the objective, not results of manuscript; Please, in the phrase “In this study, we demonstrate that 18 months…” the word “demonstrate” need correction;
6. Why the authors used only female? A complementary study with male will be interesting;
7. The authors have three groups, but I think that a group with young mice + NR improve the study;
8. The NR compound causes toxicity in young mice? Are there some reports in the literature?
9. Please provide a reference to “Mice were transcardially perfused with 4 ℃ saline”. Why not PBS?
10. Please provide information about the induction of non-alcoholic fatty liver disease, only the age caused this disease?
11. Please provide all F values, P values of data analyzed; The authors could make a table;
12. Why the authors did not use the young group in the immunohistochemical:
13. Please correct the phrase in the discussion section “The principal findings arose from the present study”.
14. The hepatic word is discussion section is “heptic”;
15. Please change “…we show the proof that…, by “ In the present study, we show…”; and “Vitamin” by “vitamin”

Reviewer 2 ·

Basic reporting

The manuscript is well-structured with a substantial content of data and based on significant background.
The English language is used clear, though some words/expression could be changed (i.e. crux (l. 69); proofs (l. 76); hepar (l. 41)).

Experimental design

The data are original. In some aspects, there are similarities (in context) with the study “Nicotinamide riboside attenuates alcohol induced liver injuries via activation of SirT1/PGC-1α/mitochondrial biosynthesis pathway” (DOI: 10.1016/j.redox.2018.04.006), however, the results are different.
A major issue, that needs to be mentioned in methods/discussion, is the use of female mice for the study. The authors should mention why only female mice were studied, and potential hormonal interference in the experimental data.

Validity of the findings

The discussion is well stated with the findings of the study. In addition, the authors also mention future goals for the study.

Additional comments

The manuscript seems to be original, it is original and exciting. However, a major issue needs to be elucidated, regarding why only female mice were used. In the current form, the manuscript should be accepted with some clarifications.

---

## Round 0.2 · accepted · Accept

· Academic Editor

Accept

I appreciate very much the effort you put into the revisions. Many of the original concerns have been addressed or at least explained.